# Prescribing Patterns and Variations of Antibiotic Use for Children in Ambulatory Care: A Nationwide Study

**DOI:** 10.3390/antibiotics11020189

**Published:** 2022-01-31

**Authors:** Githa Fungie Galistiani, Ria Benkő, Balázs Babarczy, Renáta Papp, Ágnes Hajdu, Éva Henrietta Szabó, Réka Viola, Erika Papfalvi, Ádám Visnyovszki, Mária Matuz

**Affiliations:** 1Department of Clinical Pharmacy, Faculty of Pharmacy, University of Szeged, 6725 Szeged, Hungary; Galistiani.Githa.Fungie@stud.u-szeged.hu (G.F.G.); benkoria@gmail.com (R.B.); tothne.viola.reka@szte.hu (R.V.); 2Faculty of Pharmacy, Universitas Muhammadiyah Purwokerto, Jl. KH. Ahmad Dahlan, Purwokerto 53182, Indonesia; 3Central Pharmacy Department, Albert Szent Györgyi Medical School, University of Szeged, 6725 Szeged, Hungary; 4Emergency Department, Albert Szent Györgyi Medical School, University of Szeged, 6725 Szeged, Hungary; 5National Public Health Center, 1097 Budapest, Hungary; babarczy.balazs@gmail.com (B.B.); hajdu.agnes@nnk.gov.hu (Á.H.); 6Office for the Vice-Rector for Science and Innovations, Semmelweis University, 1085 Budapest, Hungary; renata.papp@gmail.com; 7Department of Primary Health Care, University of Pécs, 7623 Pécs, Hungary; 8Department of Paediatrics and Paediatric Health Centre, Albert Szent Györgyi Medical School, University of Szeged, 6720 Szeged, Hungary; szaboeva1978@yahoo.com; 9Department of Neurosurgery, Albert Szent Györgyi Medical School, University of Szeged, 6725 Szeged, Hungary; 10Internal Medicine Clinic, Infectology Unit, Albert Szent Györgyi Medical Center, 6725 Szeged, Hungary; papfalvi.erika.piroska@med.u-szeged.hu (E.P.); visnyovszki.adam@med.u-szeged.hu (Á.V.)

**Keywords:** drug utilisation study, prescription rate, physician specialty, seasonality, regional variation, outpatient use, public health, antibiotic exposure, antibiotic stewardship, number of prescriptions

## Abstract

The aim of this study was to analyse characteristics of paediatric antibiotic use in ambulatory care in Hungary. Data on antibiotics for systemic use dispensed to children (0–19 years) were retrieved from the National Health Insurance Fund. Prescribers were categorised by age and specialty. Antibiotic use was expressed as the number of prescriptions/100 children/year or month. For quality assessment, the broad per narrow (B/N) ratio was calculated as defined by the European Surveillance of Antimicrobial Consumption (ESAC) network. Paediatric antibiotic exposure was 108.28 antibiotic prescriptions/100 children/year and was the highest in the age group 0–4 years. Sex differences had heterogenous patterns across age groups. The majority of prescriptions were issued by primary care paediatricians (PCP). The use of broad-spectrum agents dominated, co-amoxiclav alone being responsible for almost one-third of paediatric antibiotic use. Elderly physicians tended to prescribe less broad-spectrum agents. Seasonal variation was found to be substantial: antibiotic prescribing peaked in January with 16.6 prescriptions/100 children/month, while it was the lowest in July with 4 prescriptions/100 children/month. Regional variation was prominent with an increasing west to east gradient (max: 175.6, min: 63.8 prescriptions/100 children/year). The identified characteristics of paediatric antibiotic use suggest that prescribing practice should be improved.

## 1. Introduction

Antimicrobial resistance (AMR) is a major public health concern worldwide [1]. One of the key strategies to reduce AMR is the prudent use of antimicrobials [2]. The burden of infectious diseases in childhood is substantial with consequent frequent antibiotic prescribing [3].

According to the European surveillance report on antimicrobial consumption for 2017, the EU/EEA population-weighted mean consumption of systemic antibacterials in ambulatory care was 21.8 DDD/1000 inhabitants/day, ranging from 10.1 in the Netherlands to 32.0 in Spain [4], where DDD is the average daily dose for adults [5]. In Hungary, systemic antibiotic use in ambulatory care was 15.6 DDD/1000 inhabitants/day with a significantly decreasing trend during the observation period (2013–2017) [4]. During the last decade, paediatric antibiotic use data have become available from a few countries (Denmark, Finland, Germany, Greece, Italy, Netherlands, Norway, Serbia, Sweden, the UK) and range between 32.5 and 136.5 prescriptions/100 children/year [6,7,8,9,10,11,12,13]. However, the scale and patterns of paediatric antibiotic use in Hungary were found to be suboptimal previously [14]. In order to plan targeted interventions, an in-depth analysis of antibiotic use is needed, focusing on prescribers’ characteristics, and regional and seasonal variations.

The objective of this study was to provide updated and detailed information on the scale and patterns of paediatric antibiotic use in ambulatory care in Hungary in 2017. Differences across age groups and sexes, and seasonal and regional variations of paediatric antibiotic use were assessed. The influence of prescribers’ characteristics (age and specialty) on paediatric antibiotic use was also analysed.

## 2. Results

### 2.1. Scale and Patterns of Antibiotic Use

In total 6,792,714 antibiotic prescriptions were dispensed and redeemed at pharmacies in 2017 in Hungary. Almost one-third (30.54%, N = 2,074,526) of these prescriptions were issued for children (Table 1), with a prescribing rate of 108.28 antibiotic prescriptions/100 children/year.

As shown in Table 2, antibiotic exposure was the highest in the youngest age group (0–4 years: 183.9 prescriptions/100 children/year). Antibiotic exposure decreased with increasing age in childhood, but there was a slight increment in the age group of late adolescents (15–19 years).

Boys in the youngest age group (0–4 years) were exposed to substantially higher antibiotic use compared to the girls. Concerning all children (0–19 years), antibiotic exposure was slightly higher in girls, due to the higher prescribing rate for girls aged 15–19 years.

Regarding patterns of antibiotic use (Table 2), broad-spectrum penicillins, cephalosporins and macrolides were the most frequently prescribed agents (87.11 prescriptions/100 children/year of broad-spectrum vs. 16.82 and 4.35 prescriptions/100 children/year of unclassified and narrow-spectrum antibiotics, respectively). The B/N ratio was high across all age groups, and peaked in children aged 15–19 years. Similarly, the proportion of narrow-spectrum penicillins, cephalosporins and macrolides (N%, see Table 2) was the lowest in this paediatric age group.

Overall, thirty active agents were used in the paediatric population in 2017, and the DU-90 segment (the number of drugs accounting for 90% of drug use [16]) consisted of 8–11 antibacterials in various paediatric subgroups. Our study shows that almost one-third (30.81%) of all prescribed antibiotics in paediatric ambulatory care was comprised of amoxicillin and clavulanic acid (co-amoxiclav) (see Appendix A).

### 2.2. Antibiotic Use According to Prescribers’ Age and Specialty

During the study period, primary care paediatricians (PCPs) were responsible for the majority (57.5%) of paediatric antibiotic prescriptions issued in ambulatory care. As shown in Figure 1, physicians prescribed mainly broad-spectrum penicillins, cephalosporins and macrolides. On the other hand, physicians aged 65 and older tended to prescribe less broad-spectrum agents than their younger colleagues. Otolaryngologists prescribed broad-spectrum penicillins, cephalosporins and macrolides more often than other medical specialists.

### 2.3. Regional Variation of Paediatric Antibiotic Use

Both the scale and patterns of paediatric antibiotic use showed regional differences within the country. Regarding the scale of paediatric antibiotic use, an increasing west to east gradient has been detected (see Figure 2a). The highest value was 175.56 prescriptions/100 children/year (in Szabolcs-Szatmár-Bereg county), while the lowest was 63.84 prescriptions/100 children/year (in the capital, Budapest). In terms of pattern (percentage of narrow-spectrum penicillins, cephalosporins and macrolides = N%), no clear geographical gradient was found (see Figure 2b). The highest relative use of narrow-spectrum antibacterials was 19.3% detected in Vas county, whereas the lowest relative use was 1% in a south-eastern county, Békés.

### 2.4. Seasonal Variation in Paediatric Antibiotic Use

Seasonal variation was substantial (see Figure 3), with monthly prescription rate peaking in January (16.65 prescriptions/100 children/month), while the lowest rate was observed in July (3.96 prescriptions/100 children/month). Seasonal variation was detectable in all age groups, with the highest rate in the youngest ones (0–4 years) (see Figure 3).

## 3. Discussion

This is the first in-depth study on paediatric antibiotic use in ambulatory care in Hungary, covering certain characteristics of paediatric antibiotic use that are rarely evidenced in the international literature (e.g., seasonality).

### 3.1. Scale and Patterns of Antibiotic Use

Despite different age groups being analysed in other countries, it is still apparent that antibiotic use in children under 19 years of age in ambulatory care in Hungary (108.28 prescriptions/100 children/year) is relatively high compared to data from Germany (0–14 years of age: 42.8 prescriptions/100 children in 2018) [17], Finland (0–17 years of age: 37.4 prescriptions/100 children in 2016) [7] and Denmark (0–19 years of age: with 32.57 prescriptions/100 children in 2017) [6], but lower than in Serbia (0–18 years of age: 136.5 antibiotic prescriptions/100 children in 2013) [12], and Greece (0–19 years of age: the annual rate was 110 prescriptions/100 children between 2010–2013) [9]. Besides the contributing factors related to prescribers, patients and the healthcare system, a lack of clear national guidelines on a watchful waiting period for certain respiratory tract infections might explain the defensive use of antibiotics in childhood.

In our study, the highest antibiotic exposure was detected in the youngest age group (0–4 years), similarly to the cross-national study of Holstiege et al. (Denmark, Germany, Italy, the Netherlands and the UK) [10], the national survey of Blix et al. from Norway [11], and data from Danish [6] and Swedish surveillance reports [13]. Hungarian paediatric antibiotic use in the youngest age group was considerably higher (with 183.9 prescriptions/100 children/year) compared to data of Swedish (33.8 prescriptions/100 children/year) [13] and Danish children (50.7 prescriptions/100 children/year) [6].

The highest antibiotic exposure of the youngest age group may be related to the immature immune system of those 0–4 years old that renders them more susceptible to infections [18] and the related overcautiousness of doctors and caregivers. Nevertheless, the magnitude of difference in paediatric outpatient antibiotic use in the youngest age group between Hungary and the Scandinavian countries is striking and difficult to justify given that Hungarian children tend to attend nursery and kindergarten later due to the exceptionally long paid maternity leave in the country (up to 3 years in the case of singletons and up to 6 years in the case of twins). Regarding other age groups, antibiotic use decreased with the increasing age of children, but a slight increase was detected in the subgroup of 15–19-year-olds. Such an increase in antibiotic exposure in late adolescence has also been observed among Swedish [13] and Danish children [6]. The lifestyle of older adolescents (e.g., sharing food and drinks with friends, onset of sexual activity) and a consequent higher risk and burden of infections might be related to this increase in antibiotic use.

Regarding sex-specific differences, the overall prescribing rate was slightly higher for girls. However, we observed variations across the different age groups. Boys in the youngest age group (0–4 years) received significantly more outpatient antibiotic prescriptions than girls. This may be related to the higher susceptibility of boys to infections [19,20,21,22,23]. In parallel with our findings, previous studies reported that boys aged 0–4 years were prescribed more antibiotics than girls, while over the age of 4, antibiotic exposure was higher in girls [8,24]. In our study, girls received more antibiotic prescriptions compared to boys in their late adolescence (15–19 years). Similar results have been reported from Norway [11], but contrary data were reported from Italy [25]. These slight differences in antibiotic exposure could be explained by different types and incidence of infections in different age groups and across sexes. For instance, acute otitis media occur most frequently among children aged less than 2 years, and the incidence rates are slightly higher among boys [26]. On the contrary, the prevalence of symptomatic infections of the genitourinary tract is higher in adolescent girls [11].

Overall, broad-spectrum penicillins, cephalosporins and macrolides were frequently prescribed for children in Hungary, with the highest proportion of broad-spectrum agents being prescribed for the youngest age group (0–4 years). This might be explained by the frequent use of broad-spectrum penicillins (i.e., co-amoxiclav) for the treatment of acute otitis media [26]. Similar to our results, broad-spectrum antibiotics were the most commonly prescribed antibiotics for children in many other countries, for instance in Greece [9], Lithuania [27], and Serbia [12].

In Hungary, co-amoxiclav was found to be the most frequently used antibiotic in children of all age groups. Although commonly prescribed antibiotics vary by country, co-amoxiclav is often the most widely used antibacterial agent in children, e.g., in Italy (Viareggio and Emilia Romagna region) [10,28], Latvia [29], and Serbia [12]. In line with this finding, a global study on antibiotic consumption reported an increased paediatric use of co-amoxiclav between 2011 and 2015, with an annual increase rate of 6.8% in low-/middle-income countries [30]. The frequent use of co-amoxiclav in Hungary may be explained by the fact that it had been marketed earlier than amoxicillin alone (without clavulanic acid), and it is still massively promoted [31].

### 3.2. Antibiotic Prescription According to Prescribers’ Age and Specialty

PCPs were found to prescribe the majority of antibiotics for children (0–19 years). Due to a lack of recent studies regarding antibiotic prescribers’ specialty in other European countries, we can only compare our data to the results of one study from a US state (Tennessee) published in 2016. In line with our findings, this study reported that paediatricians accounted for 57% of total outpatient antibiotic prescriptions in children [32].

Another finding of our study was that physicians aged 65 and older tended to prescribe less broad-spectrum agents than their younger colleagues. This might be explained by the fact that they were also familiar with older, narrow-spectrum agents, such as phenoxymethylpenicillin (penicillin V). Overall, our study revealed that the majority of physicians prescribed more broad-spectrum antibiotics than narrow-spectrum agents. A systematic review revealed the reasons behind this phenomenon, reporting the factors that influence antibiotics prescribing. These include patients’ expectations, the severity and duration of the infection, diagnostic uncertainty, fear of potentially losing patients, and the influence of pharmaceutical companies [33]. Similarly, the lack of updated national therapeutical guidelines, the limited availability of narrow spectrum antibacterials in the Hungarian pharmaceutical market and the influence of pharmaceutical companies might in part be responsible for the observed pattern in Hungary [34], but further research is required to better understand physicians’ decision making about paediatric antibiotic prescribing.

Prescribing broad-spectrum agents was found to be common practice by all doctors, irrespective of their specialty. Otolaryngologists prescribed broad-spectrum antibacterials more often than others, probably because they treat more complicated cases.

### 3.3. Regional Variation in Paediatric Antibiotic Use

There was a more than 2.5-fold difference between the counties with the highest and the lowest paediatric antibiotic consumption. This is considerably higher than the 1.9-fold regional difference reported from Germany in 2018 [17]. The lowest paediatric prescription rate found in the capital (Budapest) and the increasing gradient from west to east might be explained by several factors, such as regional differences in infection rates, socio-economic characteristics and health literacy of patients/families, access to paediatric GPs and physicians’ workload [32,33,35]. In line with this finding, a higher annual number of consultations per PCP (higher workload) was registered in those counties that were characterised by higher prescription rates compared to the capital city [36]. Moreover, some chronic diseases leading to an increased susceptibility to infections, such as asthma and diabetes mellitus, have a higher prevalence rate among children in counties with higher antibiotic use [36]. Moreover, a recent publication on Hungarian primary healthcare availability reported that the most deprived areas were found in the north-eastern and south-western parts of Hungary, whereas the least deprived areas were in the north-western part of the country, as well as in the capital city and its neighbouring areas [37] where the lowest rates of antibiotic exposure were detected in our study.

Regarding the percentage of narrow-spectrum penicillins, cephalosporins and macrolides, an odd regional variation was found in Hungary. This may indicate a difference in prescribing habits in different counties (regions). The reason for why substantially higher proportion of narrow spectrum antibiotic consumption was found in the westernmost area of the country (Vas county) remains unclear, and needs further investigation to reveal what contributed to this more optimal prescribing practice.

### 3.4. Seasonal Variation in Paediatric Antibiotic Use

The highest monthly antibiotic prescription rates in the paediatric population were observed during the winter season, peaking in January. Seasonal fluctuation of outpatient antibiotic use in the general population across Europe has been described before [38], and it is most likely to be related to the higher incidence of viral respiratory infections in winter months [38,39]. A previous cross-national comparison study in the paediatric population showed that seasonal peaks of antibiotic exposure in winter months were most pronounced in countries with high overall antibiotic utilisation, e.g., in Italy, followed by Germany [10]. In our study, a slightly increasing rate of antibiotic use was seen from August to September, which can be explained by children attending school again after the summer holiday.

### 3.5. Strengths and Limitations

The use of a large, population-based dataset covering the whole population as well as the good drug coverage (see below in the methods) are the main strengths of our study. However, we were not able to evaluate the appropriateness of prescribing, nor were we able to analyse the correlation between diagnoses and antibiotic use in this study. One year of data might be regarded as suboptimal to report seasonal variation. However, as the year 2017 was not extraordinary in terms of antibiotic use or overall seasonality, this limitation might not affect the implications on seasonal variation [40].

## 4. Materials and Methods

### Study Design, Study Period and Data Source

This was a population-based, descriptive, retrospective study pertaining to the year 2017. Information on the use of systemic antibiotics—Anatomical Therapeutic Chemical (ATC) code J01—was obtained from the National Health Insurance Fund of Hungary (in Hungarian: Nemzeti Egészségbiztosítási Alapkezelő, acronym: NEAK), which contains information on reimbursed drugs dispensed in Hungary prescribed in ambulatory care. As almost all available antibiotics are reimbursed in Hungary, the database provides information for ~95% of total antibiotic dispensing. As NEAK is the only public health insurance fund in Hungary, the database has 100% population coverage. Reimbursed prescriptions issued by general practitioners of any kind such as PCPs (also commonly referred as paediatrician GP in the Hungarian terminology), as well as specialists and dentists, prescribed for patients in ambulatory care and patients visiting private practices are included in the dataset. A more detailed description of paediatric healthcare in Hungary is provided in a manuscript by Laszlo et al. [41].

Analysis was focused on children aged 0–19 years. Data were stratified by age group of patients (0–4 years; 5–9 years; 10–14 years; and 15–19 years), age group of physicians (< 40 years, 40–65 years, 65 < years), speciality group of physicians (e.g., PCP, otolaryngologist), region (i.e., county), and month of antibiotic dispensation. Antibiotic use was expressed as the number of prescriptions per 100 inhabitants in the age group concerned per year or per month. Population data were derived from Eurostat. The proportion of broad-spectrum penicillins, cephalosporins and macrolides, as defined by the European Centre for Disease Prevention and Control [15], was calculated.

## 5. Conclusions

High paediatric outpatient antibiotic exposure with a suboptimal pattern was detected in Hungary, especially among children in the youngest age group (0–4 years). The considerable regional variation and the disproportionately high seasonality of antibiotic use in children may also suggest suboptimal prescribing practices. The revealed characteristics of paediatric outpatient antibiotic use can support planning antibiotic stewardship interventions and can serve as a basis for more detailed qualitative research.

## Figures and Tables

**Figure 1 antibiotics-11-00189-f001:**
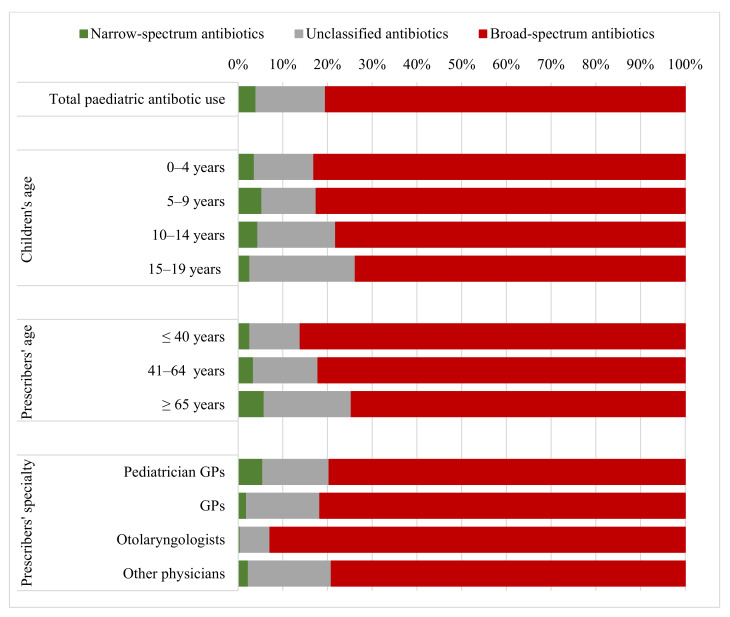
Use of antibiotics of broad-/narrow-spectrum categories defined by ESAC [15] according to different prescribers’ groups and children’s age groups in ambulatory care in Hungary, 2017. Broad-spectrum antibiotics: broad-spectrum penicillins, cephalosporins and macrolides (J01(CR+DC+DD+[F-FA01])). Narrow-spectrum antibiotics: narrow-spectrum penicillins, cephalosporins and macrolides (J01(CE+DB+FA01)). Unclassified antibiotics: all other antibiotics.

**Figure 2 antibiotics-11-00189-f002:**
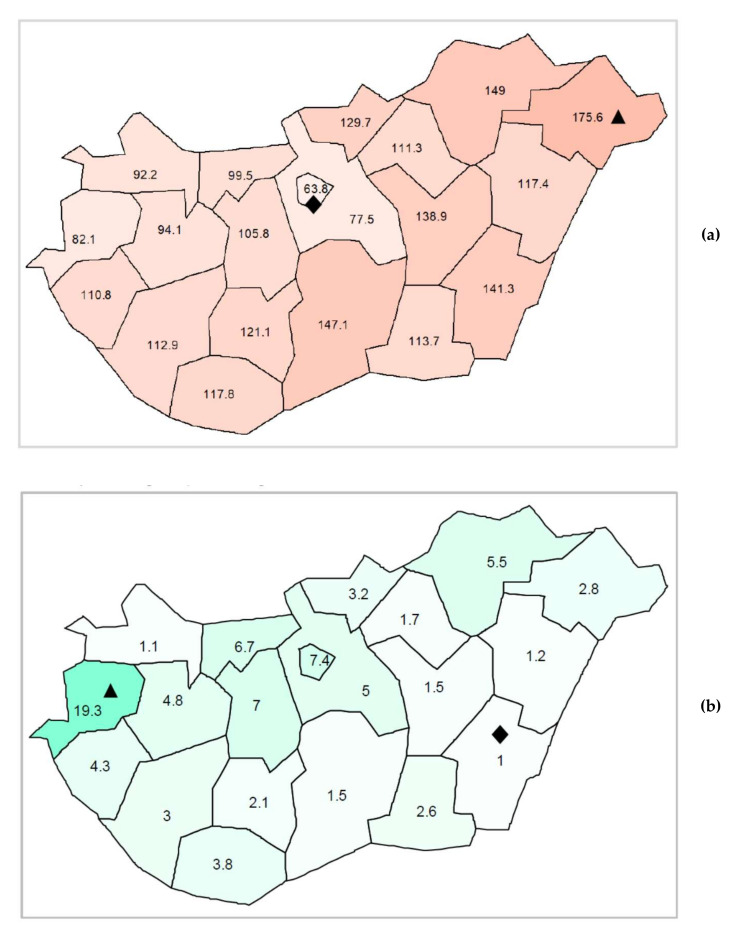
(**a**) Regional differences in paediatric antibiotic use (expressed as the number of prescriptions/100 children/year) in Hungary, 2017; Diamond symbol (♦) for the lowest prescribing rate, triangle symbol (▲) for the highest prescribing rate. (**b**) Regional differences of paediatric antibiotic use (expressed as the proportion of narrow-spectrum penicillins, cephalosporins and macrolides relative to all antibiotics) in Hungary, 2017; Diamond symbol (♦) for the lowest percentage, triangle symbol (▲) for the highest percentage.

**Figure 3 antibiotics-11-00189-f003:**
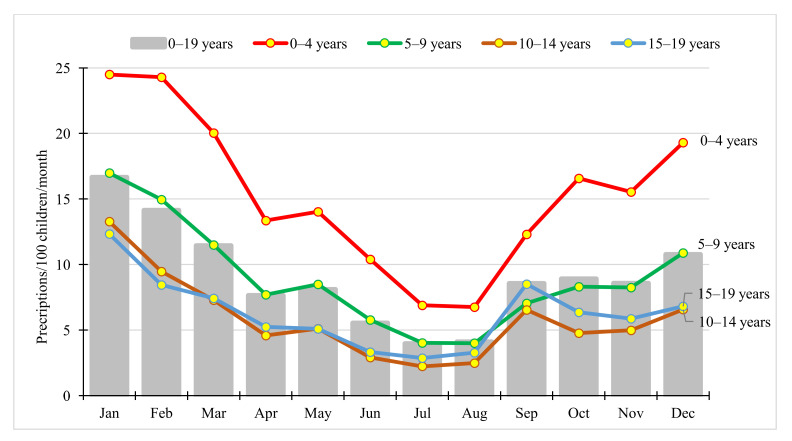
Seasonality of antibiotic use in children by age groups (expressed as the number of prescriptions/100 children/month) in Hungary, 2017.

**Table 1 antibiotics-11-00189-t001:** Antibiotic exposure in ambulatory care in Hungary, 2017.

Age Groups	Number of Population	Number of Antibiotic Prescriptions	Percentage of Total Prescriptions Redeemed (%)
0–4 years	461,739	849,139	12.50
5–9 years	474,702	511,965	7.54
10–14 years	486,424	341,209	5.02
15–19 years	493,069	372,213	5.48
All children and adolescents	1,915,934	2,074,526	30.54
All inhabitants	9,797,561	6,792,714	100.00

**Table 2 antibiotics-11-00189-t002:** Scale and characteristics of age- and sex-specific antibiotic use for children in ambulatory care in Hungary, 2017.

Sex	Age Groups (years)	Prescription/100 Children/Year				B/N	N %
		All Antibiotics	B	N	Unclassified		
All Children	0–19	108.28	87.11	4.35	16.82	20.04	4.01
Girls	0–19	109.88	87.06	4.38	18.44	19.88	3.99
Boys	0–19	106.76	87.16	4.32	15.28	20.19	4.04
All children	0–4	183.90	152.71	6.74	24.45	22.67	3.66
	5–9	107.85	89.01	5.74	13.10	15.50	5.32
	10–14	70.15	54.84	3.10	12.20	17.69	4.42
	15–19	75.49	55.70	2.00	17.79	27.86	2.65
Girls	0–4	177.13	146.30	6.54	24.29	22.35	3.69
	5–9	107.26	87.81	5.73	13.71	15.32	5.34
	10–14	71.03	54.97	3.22	12.85	17.10	4.53
	15–19	87.78	62.53	2.19	23.06	28.54	2.50
Boys	0–4	190.31	158.78	6.92	24.61	22.96	3.63
	5–9	108.41	90.14	5.75	12.52	15.68	5.30
	10–14	69.31	54.73	2.99	11.59	18.30	4.32
	15–19	63.89	49.25	1.82	12.83	27.09	2.84

Broad- and narrow-spectrum categories defined by ESAC [15]. B = broad-spectrum penicillins, cephalosporins and macrolides (J01(CR+DC+DD+[F-FA01])). N = narrow-spectrum penicillins, cephalosporins and macrolides (J01(CE+DB+FA01)). Unclassified: all other antibiotics. B/N = Ratio of the consumption of broad-spectrum penicillins, cephalosporins and macrolides to the consumption of narrow-spectrum penicillins, cephalosporins and macrolides. N% = the proportion of narrow-spectrum penicillins, cephalosporins and macrolides.

## Data Availability

Data are available from the corresponding author upon request.

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
