# Peer review of "Prescribing Patterns and Variations of Antibiotic Use for Children in Ambulatory Care: A Nationwide Study"

_antibiotics, 2022, doi:10.3390/antibiotics11020189_

Round 1
Reviewer 1 Report
This is an extensive and important study to address the topic of antibiotic use among the pediatric population, the author described the pattern and volume of pediatric use and issued the differences across age groups and sexes, seasonal and regional variations.
Here are some comments for consideration to improve the quality of the manuscript.
Introduction: (1) The author cited some surveillance consumption data to illustrate how the antibiotics were used, however, these data were population-weighted as the author stated. As the study was focusing on pediatric population, more specific numbers of antibiotic use among children is preferred.
Results: (1) Table 1 illustrated the scale and pattern of antibiotic use. The International Convention on the Rights of the Child defines a child as anyone under the age of 18. I understand that the data was retrieved from a given database. Would it be possible to set the age scale among 0-18 years old?
Materials and Methods: (1) This part of the logic is a bit confusing, I suggest to better organize this part using the STROBE checklist, which provided items that should be included in reports of observational studies. (2) Due to the prescription data were not linked with diagnoses, and the advantage of the dataset of the study which covered the whole population, my recommendation is to adopt WHO’s AWaRe classification to better describe the patterns of antibiotic use. (3) (2) Normally seasonal variation was used to observe the distribution patterns between different years. Although it make sense that winter season is associated with more antibiotic consumption, it is not that sound to draw the conclusion of seasonal variation patterns just relying on one-year data.
Author Response
Response to Reviewer 1 Comments
Reviewer comments: This is an extensive and important study to address the topic of antibiotic use among the pediatric population, the author described the pattern and volume of pediatric use and issued the differences across age groups and sexes, seasonal and regional variations. Here are some comments for consideration to improve the quality of the manuscript.
First of all, thank you for your detailed review, which helped us to improve our manuscript. Hope that you will found our answers sound.
Point 1: Introduction: (1) The author cited some surveillance consumption data to illustrate how the antibiotics were used, however, these data were population-weighted as the author stated. As the study was focusing on pediatric population, more specific numbers of antibiotic use among children is preferred.
Response 1: Thank you for suggesting this. In the old version, we only mentioned these studies (studies on pediatric antibiotic use) in the discussion. In the new version we also mentioned and cited these references in the introduction (see page 2, line 54-57).
Point 2: Results: (1) Table 1 illustrated the scale and pattern of antibiotic use. The International Convention on the Rights of the Child defines a child as anyone under the age of 18. I understand that the data was retrieved from a given database. Would it be possible to set the age scale among 0-18 years old?
Response 2: Thank you for you consideration about the range of children age. We received data stratified by 4 subgroups, each involving 5 year (0-4, 5-9, 10-14, 15-19), so we cannot change that unfortunately. As other published studies and reports also use this age-range, hopefully you can accept this. For example, report on SWEDRES-SVARM (Swedish Antibiotic Utilisation and Resistance in Human Medicine-Swedish Veterinary Antibiotic Resistance Monitoring), DANMAP (The Danish Integrated Antimicrobial Resistance Monitoring and Research Programme)-Use of antimicrobial agents and occurrence of antimicrobial resistance in bacteria from food animals, food and humans in Denmark), a paper by de Jong et. al. titled “Antibiotic drug use of children in the Netherlands from 1999 till 2005” (DOI: 10.1007/s00228-008-0479-5) also considered children from 0 to 19 years.
Point 3: Materials and Methods: (1) This part of the logic is a bit confusing, I suggest to better organize this part using the STROBE checklist, which provided items that should be included in reports of observational studies. (2) Due to the prescription data were not linked with diagnoses, and the advantage of the dataset of the study which covered the whole population, my recommendation is to adopt WHO’s AWaRe classification to better describe the patterns of antibiotic use. (3) (2) Normally seasonal variation was used to observe the distribution patterns between different years. Although it make sense that winter season is associated with more antibiotic consumption, it is not that sound to draw the conclusion of seasonal variation patterns just relying on one-year data.
Response 3: (1) Thank you for this comment. We used this structure required by the Antibiotics journal, which put the methodology section at the end (before the conclusion section). We filled in the STROBE checklist and attached as a supplementary file (it was renamed by the system to "author-coverletter-16758408.v2.doc") (2) Thank you for your recommendation to adopt WHO’s AWaRe classification. In the recent manuscript it is already too many data, figures and tables provided, we will consider to use this classification in our next study. (3) Thank you for this remark. We agree that only one-year data for the seasonal variation is a limitation, so we should phrase the implications modestly. We complemented our limitation section in the new version of manuscript. On the other hand, we checked our surveillance data that year (2017) was not an extraordinary year in terms of antibiotic use or seasonality (whole population), so we believe minor bias was introduced by focusing only on a single year data.

Reviewer 2 Report
- The Article clearly lacks the standard IMRAD format.
- As a matter of fact, the Methodology is the second last section and then directly conclusion. It is very unusual.
- Methods: It is only a record-based study, and thus doesn't add much
- The entire article needs reorganization and standard presentation. In the present form, it is not scientific.
Author Response
Response to Reviewer 2 Comments
Point 1: The Article clearly lacks the standard IMRAD format.
Point 2: As a matter of fact, the Methodology is the second last section and then directly conclusion. It is very unusual.
Point 4: The entire article needs reorganization and standard presentation. In the present form, it is not scientific.
Response 1, 2, 4 : First of all, thank you for the comments. We agree that our manuscript format is unusual and does not follow the IMRAD structure. If you check the author instruction for the journal Antibiotics (on this following link, https://www.mdpi.com/journal/antibiotics/instructions), you will see that methodology section should be after the discussion section, so the required journal style do not follow the IMRAD format.
Point 3: Methods: It is only a record-based study, and thus doesn't add much
Response 3: Thank you for noting this, but we do not share your opinion. We presented many new findings that has not been revealed before. For example, paediatric antibiotic prescribing according to prescribers’ age and specialty, data on seasonality and regional disparities. We believe that these new findings will enrich the knowledge on paediatric antibiotic use.
Reviewer 3 Report
This is a well-written nationwide study on antibiotic use in primary healthcare setting in Hungary.
Minor comment: Page 2. Line 51. Please define DDD.
Author Response
Response to Reviewer 3 Comments
Comments: This is a well-written nationwide study on antibiotic use in primary healthcare setting in Hungary.
Point 1: Minor comment: Page 2. Line 51. Please define DDD.
Response 1: Thank you for the positive remark. In the new version we added the definition of DDD as you suggested (see page 2, line 52). Hope this more clear in the new manuscript version.
Round 2
Reviewer 2 Report
Satisfied with the responses of the authors
Author Response
Thank you for your positive remarks